# Phospho-Mutant Activity Assays Provide Evidence for the Negative Regulation of Transcriptional Regulator PRE1 by Phosphorylation

**DOI:** 10.3390/ijms21239183

**Published:** 2020-12-02

**Authors:** Minmin Wang, Yanchen Tian, Chao Han, Chuanen Zhou, Ming-Yi Bai, Min Fan

**Affiliations:** The Key Laboratory of Plant Development and Environmental Adaptation Biology, Ministry of Education, School of Life Science, Shandong University, Qingdao 266237, China; sddzwangmm@163.com (M.W.); tych2211@163.com (Y.T.); hanchao@sdu.edu.cn (C.H.); czhou@sdu.edu.cn (C.Z.); baimingyi@sdu.edu.cn (M.-Y.B.)

**Keywords:** Arabidopsis, transcriptional regulator, PRE1, phosphorylation, plant biomass

## Abstract

The PACLOBUTRAZOL-RESISTANCE (PRE) gene family encodes a group of atypical helix-loop-helix (HLH) proteins that act as the major hub integrating a wide range of environmental and hormonal signals to regulate plant growth and development. PRE1, as a positive regulator of cell elongation, activates HBI1 DNA binding by sequestering its inhibitor IBH1. Furthermore, PRE1 can be phosphorylated at Ser-46 and Ser-67, but how this phosphorylation regulates the functions of PRE1 remains unclear. Here, we used a phospho-mutant activity assay to reveal that the phosphorylation at Ser-67 negatively regulates the functions of PRE1 on cell elongation. Both of mutations of serine 46, either to phospho-dead alanine or phospho-mimicking glutamic acid, had no significant effects on the functions of PRE1. However, the mutation of serine 67 to glutamic acid (*PRE1^S67E^-Ox*), but not alanine (*PRE1^S67A^-Ox*), significantly reduced the promoting effects of PRE1 on cell elongation. The mutation of Ser-67 to Glu-67 impaired the interaction of PRE1 with IBH1 and resulted in PRE1 failing to inhibit the interaction between IBH1 and HBI1, losing the ability to induce the expression of the subsequent cell elongation-related genes. Furthermore, we showed that *PRE1-Ox* and *PRE1^S67A^-Ox* both suppressed but *PRE1^S67E^-Ox* had no strong effects on the dwarf phenotypes of *IBH1-Ox*. Our study demonstrated that the PRE1 activity is negatively regulated by the phosphorylation at Ser-67.

## 1. Introduction

The basic/helix–loop–helix (bHLH) proteins are a super family of transcription factors that carry out important functions in all eukaryotic kingdoms [1,2,3,4,5]. The bHLH family proteins contain a bHLH domain, which has approximately 60 amino acids with two functionally distinctive regions —the basic region and the HLH region [6]. The basic region is located at the N-terminal end of the domain and consists of 15 amino acids with a high number of basic residues, which are involved in DNA binding [7,8]. The HLH region functions as a dimerization domain and contains two amphipathic-helices separated by a loop region of variable sequence and length [6,9]. Outside of the conserved bHLH domain, the bHLH family proteins exhibit considerable sequence divergence [7]. In plants, the bHLH family proteins play important roles in plant growth, development, and stress responses [4,10,11,12,13,14,15,16]. They are involved in a wide range of developmental processes, such as stomatal development [17,18,19], root hair formation [20,21,22,23], and floral development [24,25,26]. They are also involved in the regulation of signal transduction and in the metabolism of various hormones, including auxin [27,28], abscisic acid [29], and brassinosteroids (BRs) [28,30]. In addition, many bHLH proteins are involved in the environmental signal transduction and stress responses [31,32].

The PACLOBUTRAZOL-RESISTANT (PRE) family belongs to a subgroup of bHLH proteins that contains six homologous genes, including *PRE1/BNQ1(BANQUO1), PRE2/BNQ2, PRE3/BNQ4/ATBS1 (ACTIVATION-TAGGED bri1 SUPPRESSOR1)/TMO7 (TARGET OF MP7), PRE4/BNQ3, PRE5,* and *PRE6/KDR (KIDARI*) [28,30,33,34,35,36]. PRE1 was initially characterized as a positive regulator of gibberellin (GA) responses [34], but subsequent studies have elucidated that PRE1 functions as the major hub in the central growth circuit of plants in order to regulate the cell elongation downstream of a wide range of environmental signals and endogenous hormones, including light, temperature, brassinosteroid (BR), gibberellin (GA), auxin, and ethylene [28,32,37,38,39]. The overexpression of *PRE1* and its rice (Oryza sativa) homolog *INCREASED LAMINA INCLINATION1 (ILI1)* led to increased cell elongation in both Arabidopsis and rice [28]. The overexpression of *PRE3/ATBS1/TMO7* suppressed the dwarf phenotypes of *bri1-301* mutant [30]. The knockdown of the activities of the PRE-family members (PRE1, PRE2, PRE5, and PRE6/KIDARI) by artificial microRNA resulted in dwarfism and hyposensitivity to BR, GA, and high temperatures but a hypersensitivity to light [32,39]. PREs are essential for these hormonal and environmental signals in order to regulate cell elongation in higher plants.

PREs encode atypical HLH proteins in higher plants, which lack the basic regions for DNA binding and act as negative regulators of other bHLHs by forming inactive heterodimers. PRE1 and ILI1 have been reported to antagonize with another atypical HLH protein, ILI1 BINDING bHLH PROTEIN1 (IBH1), to control the cell elongation in Arabidopsis and rice [28]. Similarly, PRE3/ATBS1/TMO7 heterodimerizes with and inhibits four atypical HLH factors, named ATBS1-interacting factors (AIF1–AIF4) [28,30]. The overexpression of *IBH1* leads to dwarfism, and this phenotype is suppressed by the overexpression of *PRE1*, indicating that PRE1 promotes cell elongation by inactivating IBH1 [28]. PREs were also reported to interact with distinct classes of atypical HLH proteins such as PAR1/PAR2 and HFR1 [33,40]. These atypical HLH proteins further heterodimerize and inhibit the bHLH proteins with a growth-promoting activity, such as HBI1, BEEs, ACEs, or PIFs [32,40,41]. HBI1, as an bHLH protein, is a positive regulator of cell elongation, while IBH1 is a negative regulator of cell elongation. HBI1 directly binds to the promoters of downstream genes to regulate their expression, and IBH1 interacts with HBI1 to inhibit the DNA binding ability of HBI1, whereas PRE1 interacts with IBH1 to prevent its inhibition of HBI1 [32]. PREs function in the PRE-IBH1-HBI1/ACEs or PRE-PAR1/PAR2/HFR1-PIF triantagonistic HLH/bHLH system by integrating multiple hormonal and environmental signals to regulate cell elongation [42].

Protein phosphorylation is one of the most important protein post-translational modifications and governs a wide range of biological processes in plants, such as cellular metabolism, signal transduction, and responses to environmental stresses [43]. Multiple bHLH transcription factors have been reported to be regulated by phosphorylation. For example, PHYTOCHROME-INTERACTING FACTORs (PIFs) are from light-induced phosphorylation by PHOTOREGULATORY PROTEIN KINASES (PPKs) [44]. The PIF activity is inhibited by light through the sequential phosphorylation, ubiquitination, and degradation in a manner dependent on their interaction with active phytochrome photoreceptors [45]. SPEECHLESS (SPCH) is the master bHLH regulator for the production and patterning of the stomata [46]. The protein stability and function of SPCH are tightly regulated by protein phosphorylation mediated by multiple kinases, including BRASSINOSTEROID INSENSITIVE 2 (BIN2), MITOGENACTIVATED PROTEIN KINASEs 3 and 6 (MAPK3 and 6), and SUCROSE NON-FERMENTING 1-RELATED KINASE 1 (SnRK1) [18,47,48]. It has been reported that the mutation of Ser-39 and Ser-42 in the HLH domain of PRE3 to alanine impaired the transport of PRE3, indicating that phosphorylation regulates the transport of PRE3 [49]. However, it is unclear whether there are other phosphorylation sites in PREs and whether phosphorylation regulates the functions of PREs.

Here, we showed that the phospho-mimicking mutation of Ser-67 to glutamic acid weakened the promoting effects of PRE1 on cell elongation, while the phospho-dead mutation of Ser-67 to alanine had no significant effects on the functions of PRE1. The overexpression of *PRE1^S67A^* (*PRE1^S67A^-Ox*) showed increased cell elongation and light-green, narrow leaves, which is similar to that of *PRE1* overexpression (*PRE1-Ox*) transgenic plants, while the overexpression of *PRE1^S67E^* (*PRE1^S67^^E^-Ox*) displayed a slightly increased cell elongation. *PRE1-Ox* and *PRE1^S67A^-Ox* both suppressed but *PRE1^S67E^-Ox* had no strongly effects on the dwarf phenotypes of *IBH1-Ox*. Furthermore, we found that the mutation of Ser-67 to Glu67 significantly reduced the interaction between IBH1 and PRE1; subsequently, PRE1^S67E^ failed to impair the interaction between IBH1 and HBI1, finally losing the ability to induce the expression of cell elongation-related genes. Our study demonstrates that PRE1 activity is negatively regulated by phosphorylation at Ser-67.

## 2. Results

### 2.1. Phosphorylation at Ser-67 Negatively Regulated the Functions of PRE1

Previous studies have shown PREs as a subgroup of bHLH transcription factors that play important roles in cell elongation in response to BR, GA, temperature, and light [28,32,37,38,39]. To explore the function of PRE1 in regulating cell elongation, we conducted a phosphorylation modification survey of PREs in the published phosphorylation datasets [50]. PRE1 was found to be phosphorylated at two conserved serines, Ser-46 and Ser-67 (Figure 1A). To understand the effects of these two phosphorylation sites on the function of PRE1, Ser-46 and Ser-67 of PRE1 were mutated to glutamic acid to mimic phosphorylation at the serine residue (PRE1^S46E^ and PRE1^S67E^) or to alanine to prevent potential phosphorylation (PRE1^S46A^ and PRE1^S67A^). We then generated transgenic Arabidopsis plants expressing *PRE1, PRE1^S46A^, PRE1^S46E^, PRE1^S67A^,* and *PRE1^S67A^* driven by the constitutive *35S* promoter. Nearly 100 T1 plants for each construct were used to analyze the growth phenotypes. According to the petiole length of the fifth leaf, we classified the transgenic plants into four categories, as follows: similar to wild type (no phenotype), weak phenotype, middle phenotype, and strong phenotype. As shown in Figure 1B,C, mutations of Ser-46 to Ala-46 or Glu-46 had no effects on the phenotypes caused by the overexpression of *PRE1*, indicating that Ser-46 of PRE1 is not important for PRE1 function. The phenotype of the mutation of Ser-67 to Ala-67 is similar to that of the *PRE1* overexpression. However, the overexpression of PRE1^S67E^ reduced the ratio of plants showing long petiole phenotypes and increased the ratio with no phenotype. These results indicate that phosphorylation at Ser-67 is very important for the function of PRE1.

To determine the effects of Ser-67-mediated phosphorylation on the functions of PRE1, the transgenic plants of *PRE1-Ox, PRE1^S67A^-Ox*, and *PRE1^S67E^-Ox* with similar protein levels of PRE1, PRE1^S67A^, and PRE1^S67E^ were selected for systemic analysis of the growth of the phenotype (Figure 2A and Appendix A). The overexpression of *PRE1* and *PRE1^S67A^* both showed a longer hypocotyl and longer root compared with the wild type, whereas the overexpression of *PRE1^S67E^* displayed normal growth phenotypes similar to wild-type plants (Figure 2B,C). *EXPANSIN1* (*EXP1*) and *EXPANSIN8* (*EXP8*) are two genes encoding the cell wall proteins that loosen the cell wall [51]. The quantitative RT-PCR analysis showed that the expression levels of *EXP1* and *EXP8* were increased in *PRE1-Ox* and *PRE1^S67A^-Ox* plants compared with that in the wild-type plants but were less increased in *PRE1^S67E^-Ox* (Figure 2D). Together, these results indicate that the phosphorylation at the Ser-67 of PRE1 negatively regulates its functions.

### 2.2. Mutation of the 67th Serine to Glutamic Acid Reduced the Binding Ability of PRE1 to IBH1

Previous studies have shown that PRE1 interacted with IBH1 to release the DNA-binding ability of HBI1 and to regulate the expression of the genes that are related to cell elongation downstream of multiple hormonal and environmental signals [32]. To determine whether the phosphorylation of PRE1 regulates the interaction between PRE1 and IBH1, we first analyzed the interaction between IBH1 with PRE1, PRE1^S46A^, PRE1^S46E^, PRE1^S67A^, and PRE1^S67E^ using the yeast two-hybrid assays. Consistent with previous reports, IBH1 interacted with PRE1 in yeast. The mutation of Ser-46 of PRE1 to Ala-46 or Glu-46 and Ser-67 of PRE1 to Ala-67 all had no significant effects on the interaction between PRE1 and IBH1, whereas the mutation of Ser-67 of PRE1 to Glu-67 significantly reduced the interaction between PRE1 and IBH1 (Figure 3A,B). The in vitro pull-down assays showed that maltose-binding protein (MBP)-PRE1, MBP-PRE1^S67A^, and MBP-PRE1^S67E^ were specifically pulled down by glutathione S-transferase (GST)-IBH1, but the pull-down efficient ratio of MBP-PRE1^S67E^ was evidently lower than that of MBP-PRE1^S67A^ (Figure 3C,D). A newly developed Ratiometric Bimolecular Fluorescence Complementation (rBiFC) assay was performed to confirm the interaction between IBH1 and the mutations of PRE1 in vivo. PRE1 or PRE1^S67E^ and IBH1 were synchronously cloned into a single vector backbone, including a monomeric red fluorescent protein as an internal marker for the expression control and ratiometric analysis. In agreement with previous results, PRE1 interacted with IBH1 in the epidermal cells of the tobacco leaves, but the co-expression of PRE1^S67E^ and IBH1 resulted in very weak signals (Figure 3E,F). We further tested the interaction between PRE1 or PRE1^S67E^ and IBH1 in plants using coimmunoprecipitation assays. The results showed that PRE1 interacted with IBH1 in plants, while the mutation of Ser-67 to Glu-67 significantly reduced the interaction of PRE1 with IBH1 (Figure 3G). Together, these results strongly demonstrated that the phosphorylation at Ser-67 of PRE1 inhibited its interaction with IBH1.

PRE1 belongs to an HLH subfamily of transcription factors and has six homologous genes in Arabidopsis. Ser-67 is a conserved serine among the PRE1 family proteins. We tested whether the phosphorylation at Ser-67 of PRE1 is involved in the interaction between the IBH1 and PRE1 family proteins. PRE2 is a homolog of PRE1 and regulates the gibberellin-dependent responses in *Arabidopsis thaliana* [34]. The protein sequences analysis indicated that the Ser-68 in PRE2 is equivalent to the conserved residue Ser-67 in the PRE1 protein (Figure 1A). We performed the site-directed mutagenesis and generated the constructs, such as PRE2^S68A^ and PRE2^S68E^. The yeast two-hybrid (Y2H) assays showed that IBH1 interacted with PRE2, PRE2^S68A^, and PRE2^S68E^, but the interaction between IBH1 and PRE2^S68E^ was slight weaker than the interaction between IBH1 and PRE2 and IBH1 and PRE2^S68A^ in yeast (Appendix A). In addition, we found the interaction between IBH1 and PRE2^S68E^ was significantly weaker than the interaction between IBH1 and PRE2 and IBH1 and PRE2^S68A^ in β-gal activity analysis (Appendix A). The Bimolecular Fluorescence Complementation (BiFC) assays showed that IBH1 interacted with PRE2 and PRE2^S68A^ but not PRE2^S68E^ in plants (Appendix A). ILI1 is the homologous gene of PRE1 in rice and has been reported to interact with rice IBH1 to regulate the rice leaf angle. The Ser-79 of ILI1 corresponds to the Ser-67 of PRE1. Our results show that ILI1 and ILI1^S79A^ interacted with OsIBH1, but ILI1^S79E^ failed to interact with OsIBH1 (Appendix A). It has been reported that PRE1 also interacts with other HLH transcription factors, including AIFs and PAR1/PAR2. To test whether the phosphorylation at the Ser-67 of PRE1 regulates its interactions with AIF1 and PAR1, we performed a Y2H assay to test the interactions of PRE1 with AIF1 and PAR1. The results showed that PRE1 interacted with AIF1 and PAR2, but the mutagenesis of Ser-67 to Glu-67 significantly reduced the interaction between PRE1 and AIF1 but not PAR2 (Appendix A). These results indicate that the phosphorylation of PRE1 at Ser-67 to inhibit the interaction between PRE1 and its partners is a conserved regulator mechanism of PRE1 in higher plants.

### 2.3. PRE1^S67E^ Failed to Abolish the Interaction between IBH1 and HBI1

Previous studies have shown that PRE1 interacts with IBH1 to prevent the interaction between IBH1 and HBI1, which directly induces the expression of the genes encoding the cell wall-loosening enzymes [32]. To test whether PRE1^S67E^ regulates the interaction between IBH1 and HBI1, we performed a Yeast Three-Hybrid (Y3H) assay. The result of Y3H showed that PRE1 and PRE1^S67A^ inhibited the interaction between IBH1 and HBI1, whereas the PRE1 mutations of Ser-67 to Glu-67 failed to abolished the interaction between IBH1 and HBI1 (Figure 4A). The in vitro pull-down assays showed that the GST-IBH1-binding ability to MBP-HBI1 was inhibited by MBP-PRE1 and MBP-PRE1^S67A^ but not MBP alone or MBP-PRE1^S67E^ (Figure 4B). These results indicate that the phosphorylation of PRE1 at Ser-67 loses its ability to prevent the interaction between IBH1 and HBI1.

### 2.4. PRE1^S67E^ Failed to Suppress the Dwarf Phenotypes of IBH1-Ox

PRE1 and IBH1 function antagonistically in regulating cell elongation. The *PRE1*-overexpression transgenic plants displayed the elongated hypocotyl, light-green, and narrow leaves, whereas the *IBH1*-overexpression transgenic plants exhibited shorter hypocotyl, dark-green, and round leaves. The *PRE1-Ox/IBH1-Ox* plants showed a long hypocotyl phenotype similar to that of the *PRE1-Ox* transgenic plants. To determine the effects of phosphorylation at Ser-67 on the functions of PRE1, we analyzed the phenotypes of *IBH1-Ox* in the different genotype backgrounds, including wild type, *PRE1-Ox*, *PRE1^S67A^-Ox* and *PRE1^S67E^-Ox*. The results show that the *PRE1^S67A^*-overexpression transgenic plants displayed an elongated hypocotyl compared with the wild type, whereas for the hypocotyl length of the *PRE1^S67E^*-overexpression transgenic plants, there is no evident difference with that of the wild type. The F1 plants of *PRE1^S67A^-Ox/IBH1-Ox* exhibited a long hypocotyl phenotype similar to that of the *PRE1-Ox/IBH1-Ox* transgenic plants. However, the F1 plants of *PRE1^S67E^-Ox/IBH1-Ox* still exhibited a slightly longer hypocotyl phenotype (Figure 5A,B and Appendix A). The quantitative RT-PCR analysis showed that the expression levels of *EXP1* and *EXP8* were increased in the *IBH1-Ox/PRE1-Ox* and *IBH1-Ox/PRE1^S67A^-Ox* plants compared with that in the wild-type plants but were less increased in *IBH1-Ox/PRE1^S67E^-Ox* (Figure 5C). These results indicated that the mutation of serine to glutamic acid not only affects the interaction between PRE1 and IBH1 but also fails to suppress the dwarf phenotypes of *IBH1-Ox*.

### 2.5. Overexpression of PRE1^S67E^ Increase the Biomass of the Plant

The overexpression of *PRE1* promoted the elongation of the hypocotyl, but the *PRE1-Ox* transgenic plant displayed elongated petioles, light-green, and narrow leaves, and a significantly reduced biomass. Therefore, the over expression of *PRE1* did not actually play a promoting role in the biomass of the plants. The phenotype of the overexpression of *PRE1^S67A^* was similar with that of *PRE1-Ox*. However, we found that the phenotype of the overexpression of *PRE1^S67E^* resulted in slightly longer petioles and round leaves, as with the wild type (Col-0; Figure 6A). To further understand the biological production of these transgenic plants, we studied the fresh and dry weights of the transgenic plants that grew in the soil for weeks. The results showed that the fresh weight and dry weights of *PRE1^67E^-Ox* were evidently higher than the wild type but that of *PRE1-Ox* and *PRE1^67A^-Ox* were lower than the wild type (Figure 6B,C). This result indicates that the overexpression of *PRE1^S67E^* was significantly increased in the biomass. Figure 6D shows that the leaves of *PRE1^67E^-Ox* were both larger and more than those of the wild type, *PRE1-Ox,* and *PRE1^67A^-Ox*. In order to obtain more accurate results, we analyzed the leaf area of the fifth/sixth and seventh/eighth true leaves. The results show that the leaf areas of the fifth/sixth and seventh/eighth true leaves of *PRE1^67E^-Ox* were significantly higher than those of the wild-type models, while those of *PRE1-Ox* and *PRE1^67A^-Ox* were significantly lower than those of the wild type (Figure 6E,F). These results indicate that the overexpression of *PRE1^S67E^* is very useful for increasing crop yields.

## 3. Discussion

Protein phosphorylation is a widespread protein modification, and it regulates a wide range of cellular processes in various organisms. PREs function as a central hub integrating multiple environmental and hormonal signals to regulate plant growth and development. In this study, we demonstrate, through several lines of evidence, that phosphorylation at Ser-67 of PRE1 plays critical regulatory roles for the functions of PRE1. First, the phosphorylation datasets show that there are two phosphorylation target residues, Ser-46 and Ser-67, in the PRE1 protein. The protein alignment showed that these two serine residues are conserved in the PRE family proteins of rice and Arabidopsis. Second, phospho-dead and phospho-mimicking mutants of two serine sites showed that the overexpression of *PRE1^S46A^, PRE1^S46E^,* and *PRE1^S67A^* caused increased petioles, light-green leaves, and reduced biomass, which are similar to the phenotypes of the *PRE1-Ox* transgenic plants. While the overexpression of *PRE1^S67E^* displayed a slightly increased petiole and significantly increased the biomass. Third, the phospho-mimicking mutant of Ser-67 to Glu-67 significantly reduced the interaction between PRE1 and IBH1 and failed to impair the interaction between IBH1 and HBI1. Finally, the overexpression of *PRE1* and *PRE1^S67A^* suppressed the dwarf phenotypes of the *IBH1-Ox* plants, but the overexpression of *PRE1^S67E^* had very weak effects on the short hypocotyl caused by the overexpression of *IBH1*. Together, these results demonstrated that the phosphorylation at Ser-67 negatively regulated the functions of PRE1.

TMO7, also named PRE3 and ATBS1, is expressed in the early embryo in Arabidopsis and is transported from the proembryo toward the uppermost suspensor cell so as to specify this cell as hypophysis and to establish the primary root meristem [36,49]. In the primary root, *TMO7* is strongly expressed in the quiescent center and is absent from the columella root, but the TMO7 proteins move from the quiescent center to the columella cells. A recent study showed that phosphorylation at Ser-39 and Ser-42 regulated the movement of TMO7 [49]. *proTMO7:TMO7-GFP* showed the fluorescent signals of TMO7-GFP in the quiescent center and columella cells. However, the *proTMO7:TMO7^S39A^-GFP* and *proTMO7:TMO7^S42A^-GFP* transgenic plants displayed the fluorescent signals of TMO7^S39A^-GFP and TMO7^S42A^-GFP only in the quiescent center but not in the columella cells, suggesting that the phosphorylation at these two serine residues is important for TMO7 transport. In addition, replacement in region 8 of nine amino acids of TMO7 with a polyalanine linker with the same length also impaired the movement of TMO7. Interestingly, region 8 is conserved in PRE family proteins and contains Ser-67. These results indicate phosphorylation at Ser-67 might control the movement of TMO7, but it needs further study.

PREs are positive regulators of cell elongation downstream of a wide range of environmental and hormonal signals. Here, we showed that phosphorylation at Ser-67 negatively regulates the functions of PRE1, so we speculate that the kinase that phosphorylates PREs at Ser-67 should be a negative regulator of cell elongation. BR-INSENSITIVE (BIN2) encodes a GLYCOGEN SYNTHASE KINASE 3 (GSK3)-like kinase and is a negative regulator in the BR-signaling pathway for inhibiting cell elongation. BIN2 has been reported to phosphorylate a number of substrates that regulate plant growth and development, such as BZR1, BES1, PIF4, ARF2, AIF4, and SPCH [30,47,52,53,54,55,56]. PREs are also reported to positively regulate the BR-signaling pathway, and BIN2 is our speculated ideal kinase for the phosphorylation of PREs. Indeed, the overlay assays showed that BIN2 interacted with PRE1 in vitro, and the kinase assay showed that BIN2 phosphorylated MBP-PRE1 but not MBP-only (Appendix A). However, the mutation of Ser-67 had no significant effects on the phosphorylation of PRE1 by BIN2. These results indicate that BIN2 may not be the kinase that phosphorylates PRE1 at Ser-67. Works screening and identifying the kinases that phosphorylate PREs at Ser-67 will help us to better understand the functions of PREs.

## 4. Materials and Methods

### 4.1. Plant Materials and Growth Conditions

*Arabidopsis thaliana* ecotype Columbia (Col-0) was used as the wild-type control and as the genetic background of the transgenic plants. Plants used in this study were grown in a greenhouse at 22 ± 2°C, 60–70% relative humidity, and 150 μmol·m^−2^·s^−1^ light intensity under a 16-h light/8-h dark photoperiod for general growth and seed harvesting. These include *PRE1-Ox*, *PRE1^S67A^-Ox*, *PRE1^S67E^-Ox*, and *IBH1-Ox*. Seedlings were photocopied for measurement of hypocotyl length, root length, and leaf area by ImageJ software (ImageJ 1.48v, National Institutes of Health, Rockville, MD, USA).

### 4.2. Vector Construction

The full-length coding sequences of *PRE1*, *PRE2*, *IBH1*, *AIF1*, *PAR2, BIN2, ILI1,* and *OsIBH1* without stop codons were obtained by PCR from Arabidopsis and rice leaf cDNA, respectively, and cloned to the pENTR^TM^/SD/D-TOPO^TM^ vectors (Thermo Fisher, Waltham, MA, USA) and then subcloned subsequently into the destination vectors, including pX-YFP (p35S:C-YFP), pX-nYFP (p35S:C-nYFP), pX-cYFP (p35S:C-cYFP), pDEST15(N-GST), pBridge (GAL4BD-X-pMET25-Y), pGCBDT7 (GAL4BD-X), pGCADT7(GAL4AD-X), and pMAL2CGW (N-MBP). All the mutants of *PRE1*, *PRE2, ILI1*, and *BIN2* including *PRE1^S67A^*, *PRE1^S67E^*, *PRE1^S46A^*, *PRE1^S46E^*, *PRE2^S68A^*, *PRE2^S68E^*, *ILI1^S79A^, ILI1^S79E^,* and *BIN2^M115A^*, were obtained using a Quick-change Site-directed Mutagenesis kit (Stratagene, La Jolla, CA, USA) cloned into pENTR^TM^/SD/D-TOPO^TM^ vectors and then recombined into the indicated destination vectors. Primer sequences used for cloning are listed in Appendix A. All the vectors were introduced into *Agrobacterium tumefaciens (strain GV3101)* and transformed into Col-0 ecotype plants by the floral dip method.

### 4.3. Quantitative Real-Time PCR Analysis

Total RNA was extracted from the seedlings of wild-type or transgenic plants using Trizol reagent (TransGen, Beijing, China). One microgram of RNA was primed with oligo(dT) and reverse-transcribed using RevertAid reverse transcriptase (Thermo Fisher, Waltham, MA, USA) and used as RT-PCR templates. The real-time (RT)-PCR was performed with a gene-specific primer (Appendix A) using a CFX connect real-time PCR detection system (Bio-Rad, Hercules, CA, USA), as previously described [57]. The *PP2A* was used as an internal control, and the standard deviations were calculated based on three biological replicates. Significant differences between means were examined by Student’s *t*-test (*p* < 0.05).

### 4.4. Yeast Two-hybrid assays

The full-length cDNA of *IBH1/AIF1/PAR2/OsIBH1* were separately fused into the pBD-GAL4 vector, and *PRE1/PRE1^S67A^/PRE1^S67E^/PRE2/PRE2^S68A^/PRE2^S68E^/ILI1/ILI1^S79A^/ILI1^S79^^E^* were separately fused into the pAD–GAL4 vectors (Stratagene, La Jolla, CA, USA) and then transformed into yeast strain AH109. Yeast clones containing the BD prey and AD bait were plated on medium SD (-Trp-Leu) and cultured, then screened for growth on medium SD (-Trp-Leu-His) but with different concentrations of 3-aminotriazol for 3 days at 30 °C for the interaction assay. The β-galactosidase activity was determined using the o-nitrophenyl-β -D-galactopyranoside (ONPG) reaction and measured by a spectrophotometer.

### 4.5. β-gal Activity Analysis

The indicated plasmids were transformed into yeast strain AH109 using the yeast transformation method. These yeast strains were grown on the Leu^−^/Trp^−^ dropout solid medium for three days under 30 °C. Liquid assay for β-Gal activity using (o-nitrophenyl-β -D-galactopyranoside) ONPG (Sigma Aldrich, St. Louis, MO, USA) was performed as previously described [58]. The solution only with the indicated buffer was used as negative control, and this assay was performed with three biological replicates. Significance of differences between the experiment group and the control group was determined using a Student’s *t*-test (*p* < 0.05).

### 4.6. Yeast Three-Hybrid Assays

The full-length cDNA of *HBI1* was cloned into the pGCADT7 vector, and the coding sequences of *IBH1*, *PRE1*, *PRE1^S67A^*, and *PRE1^S67E^* were cloned into the pBridge vector, respectively. IBH1 was cloned into the Multiple Cloning Sites I (MCSI) of the pBridge vector. The three additional gene sequences of *PRE1*, *PRE1^S67A^*, and *PRE1^S67E^* were cloned into the Multiple Cloning Sites II (MCSII)of the pBridge vector. PRE1, PRE1^S67A^, and PRE1^S67E^ as the third protein is controlled by a conditional methionine promoter (pMet25), such that it is expressed in the absence of methionine. These vectors were transformed into yeast strain AH109 and grown on the selective solid medium without Leu^−^/Trp^−^/Met^−^ supplements under 30 °C for three days. This assay was performed following the manufacturer’s instructions (Clontech, Mountain View, CA, USA).

### 4.7. Bimolecular Fluorescence Complementation (BiFC) Assays

Full-length cDNA of *IBH1* was cloned into the pX–cYFP and that of *PRE1/PRE1^S67A^/PRE1^S67E^/PRE2/PRE2^S68A^/PRE2^S68E^* were separately cloned into the pX–nYFP. Agrobacterial suspensions containing IBH1-cYFP and PRE1, PRE1^S67A^, PRE1^S67E^, PRE2, PRE2^S68A^, or PRE2^S68E^-nYFP were injected into tobacco leaves from lower epidermis. The transfected tobacco plants were kept in 22 °C darkness for at least 36 h in the greenhouse so that the transfected DNA could be expressed. Fluorescent signals were detected by confocal laser scanning (Zeiss, Oberkochen, Germany).

### 4.8. Ratiometric Bimolecular Fluorescence Complementation (rBiFC) Assays

The coding domain sequence of *IBH1* and *PRE1*, *PRE1^S67E^*, were cloned into pDONR221-P1P4 and pDONR221-P3P2 using the attB×attP (BP) recombination reaction (Invitrogen, Carlsbad, CA, USA), respectively. The attL×attR (LR) recombination reaction was performed with pDONR221-P1P4-IBH1 and pBiFCt-2in1-CC firstly; then, the second LR recombination reaction was performed with pDONR221-P3P2-PRE1 or pDONR221-P3P2-PRE1^S67E^ and pBiFCt-2in1-IBH1-LacZ. Agrobacterial suspensions containing pBiFCt-2in1-IBH1-LacZ, pBiFCt-2in1-IBH1-PRE1, and pBiFCt-2in1-IBH1-PRE1^S67E^ were transformed into tobacco (*Nicotiana tabacum*) leaves. After 40-h incubation at 22 °C, the fluorescent images were visualized using an LSM700 laser scanning confocal microscope (Zeiss, Oberkochen, Germany). The fluorescence intensity (*n* = 50) was determined by ImageJ software.

### 4.9. In Vitro Pull-down Assays

The recombinant GST-fused IBH1 and MBP-fused PRE1, PRE1^S67A^, PRE1S^67E^, and HBI1 were overexpressed using bacteria. These proteins were purified using glutathione beads (GE Healthcare, Little Chalfont, Buckinghamshire, UK) and amylose resin beads (NEB), respectively. GST beads containing 1-μg GST-IBH1 were incubated with 1-μg MBP, MBP-PRE1, MBP- PRE1^S67A^, PRE1^S67E^, and/or MBP-HBI1, as indicated in the pull-down buffer (20-mM Tris-HCl, pH 7.5, 100-mM NaCl, and 1-mM EDTA), and the beads were washed 6 times with wash buffer (20-mM Tris-HCl, pH7.5, 300-mM NaCl, 0.5% (*v/v*) TritonX-100, and 1-mM EDTA). The proteins were eluted from beads using 70 μL 2× SDS loading buffer. The input and eluate were separated on 8% SDS-PAGE gels, and specific proteins were detected by immunoblot analysis using anti-MBP (NEB, Cat: E8038L, 1:5000 dilution).

### 4.10. Coimmunoprecipitation Assays

Ten-day-old *35Spro:IBH1-Myc*, *35Spro:PRE1-YFP/35Spro:IBH1-Myc*, and *35Spro:PRE1^S67E^-YFP/35Spro:IBH1-Myc* were harvested and ground to fine powder. The samples were extracted with NEB buffer (20-mM HEPES-KOH, at pH 7.5, 40-mM KCl, 1-mM EDTA, 0.5% Triton X-100, 1-mM Phenylmethanesulfonyl fluoride (PMSF), and 1× protease inhibitor cocktails; Sigma) and incubated with GFP-Trap agarose beads (Chromotek) for 2 h in a cold room; then, the beads were washed by wash buffer (20-mM HEPES-KOH, at pH 7.5, 40-mM KCl, 1-mM EDTA, 0.05% Triton X-100, and 0.1% NP40). The proteins were eluted from the beads by 2× SDS sample buffer, analyzed by SDS-PAGE, and immunoblotted with anti-YFP (TransGen Biotech, Beijing, China, Cat: N20610,1:5000 dilution) and anti-Myc (Sigma Aldrich, St. Louis, MO, USA, Cat: M4439, 1:5000 dilution) antibodies.

### 4.11. In Vitro Kinase Assays

MBP-PRE1, MBP-PRE1^S67A^, MBP-PRE1^S67E^, GST-BIN2, and GST-BIN2^M115A^ proteins were expressed and purified from *Escherichia coli* (*E. Coli*). MBP, MBP-PRE1, MBP-PRE1^S67A^, MBP-PRE1^S67E^ were incubated with GST-BIN2 and GST-BIN2^M115A^, as indicated in the kinase buffer (20-mM Tris, pH7.5, 1-mM MgCl2, 100-mM NaCl, 1-mM Dithiothreitol (DTT), and 10-μM cold ATP) containing (γ-32P)ATP (10 μCi) for 3 h at 30°C. The reaction was stopped by the addition of 10 μL of 4 × SDS loading buffer. Proteins were resolved by 10% SDS–PAGE.

## 5. Conclusions

In summary, we found the conserved Ser-67 play critical roles for the functions of PRE family proteins. Phospho-mimicking mutation of Ser-67 to Glu-67 impaired the interaction of PRE1 with IBH1 and resulted in PRE1 failing to inhibit the interaction between IBH1 and HBI1, subsequently losing the ability to induce the expression of cell elongation-related genes. *PRE1-Ox* and *PRE1^S67A^-Ox* can suppress the dwarf phenotype of *IBH1-Ox*, while *PRE1^S67E^-Ox* had no significant effects on the dwarf phenotypes of *IBH1-Ox*. These results demonstrated that phosphorylation at Ser-67 negatively regulates the functions of PRE1 on cell elongation by weakening the interaction between PRE1 and IBH1.

## Figures and Tables

**Figure 1 ijms-21-09183-f001:**
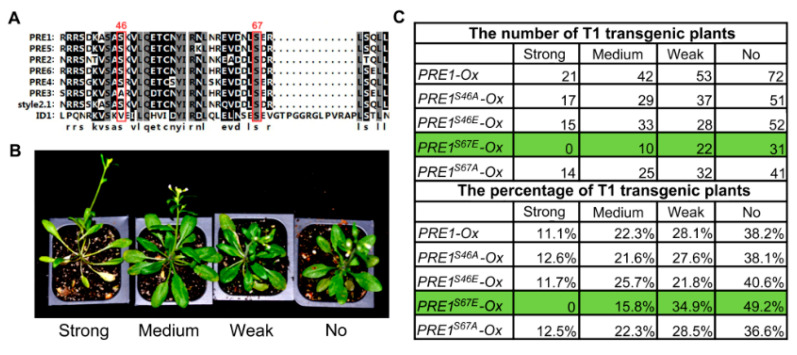
Mutation the Ser-67 of PACLOBUTRAZOL-RESISTANT-1 (PRE1) to Glu-67 reduced the promoting effects of PRE1 on cell elongation. (**A**) Protein sequences comparison within the PRE family and the Style2.1 and ID1 proteins. The alignment image was made by CLUSTAL OMEGA software. The Ser-46 and Ser-67 of PRE1 are conserved sites, labeled with a red rectangle. (**B**) Various degrees of long petiole phenotypes were observed among T1 plants overexpressing *PRE1* or mutant versions of *PRE1,* which were grown in soil for 4 weeks under long-day conditions. (**C**) The number (the upper part of the table) and percentage (the bottom part of the table) of each category of phenotypic severity among the T1 transgenic plant populations with an overexpression of *PRE1* or mutant versions of *PRE1.*

**Figure 2 ijms-21-09183-f002:**
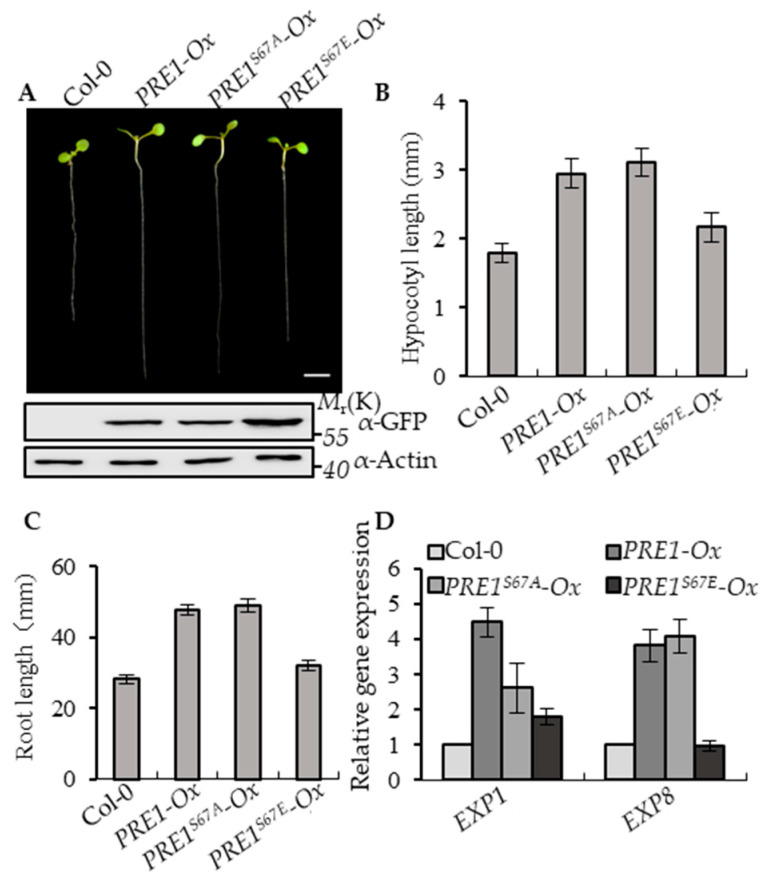
Phospho-mimicking mutation of Ser-67 to Glu-67 attenuates PRE1-induced cell elongation. (**A**) Representative plants of the wild type and overexpression of the *PRE1, PRE1^S67A^*, and *PRE1^S67E^* transgenic plants that were grown on half-strength Murashige and Skoog (MS) medium for 7 days. Scale bar = 5 mm. The bottom images show the protein levels of PRE1 and mPRE1 in Col-0 and different transgenic seedlings using the anti-green fluorescence protein (GFP) antibody and anti-actin for the loading control. (**B**) The average hypocotyl lengths and (**C**) root lengths of the wild type, *PRE1-Ox, PRE1^S67A^-Ox,* and *PRE1^S67E^-Ox* were measured from at least 20 plants. Error bars indicate standard deviation (SD; *n* = 3). (**D**) Quantitative RT-PCR analyses of the gene expressions of *EXP1* and *EXP8* in wild-type, *PRE1-Ox, PRE1^S67A^-Ox,* and *PRE1^S67E^-Ox* plants. *PP2A* was used as the internal control. Error bars are the SD from three biologic replicates.

**Figure 3 ijms-21-09183-f003:**
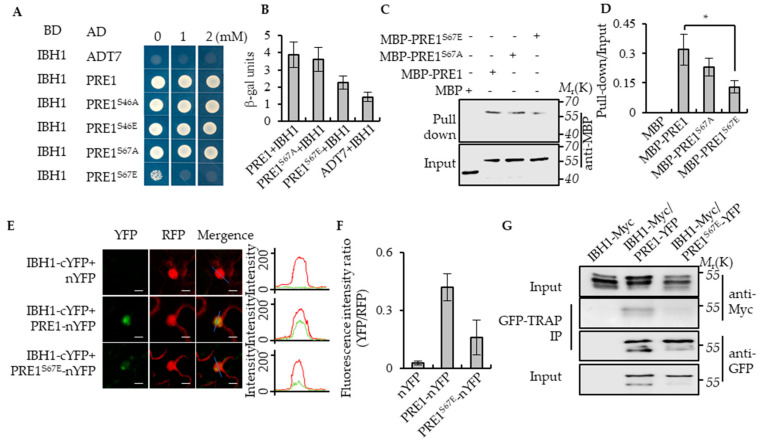
Phospho-mimicking mutation of Ser-67 to Glu-67 inhibited the interaction between PRE1 and IBH1. (**A**) Yeast two-hybrid assays indicated that IBH1 interacts with PRE1, PRE1^S46A^, PRE1^S46E^, and PRE1^S67A^ but not PRE1^S67E^. ADT7 is the GAL4AD protein used as the negative control. (**B**) Quantitative analysis of the β-galactosidase expression was induced by the interaction between BD-IBH1 and AD-PRE1s in the yeast. The mutation of Ser-67 of PRE1 to Glu-67 significantly reduced the interaction between PRE1 and IBH1. The β-galactosidase activity was measured by a spectrophotometer. Error bars indicate the SD from three biological repeats. (**C**,**D**) Protein pull-down assays indicate that the serine-67 mutation to glutamic acid reduced the PRE1-binding ability to IBH1. Quantification of the pull-down assays (normalized to input) display that the pull-down efficient ratio of MBP-PRE1^S67E^ is evidently lower than that of MBP-PRE1 and MBP-PRE1^S67A^ in vitro (**D**). Error bars indicate the SD from three biological repeats. * indicates *p* < 0.05. (**E**,**F**) Ratiometric-Bimolecular Fluorescence Complementation (rBiFC) assays show that the mutation of serine-67 to glutamic acid reduces the interaction between PRE1 and IBH1 in tobacco. The fluorescent intensity of YFP (interaction signal) and RFP (constitutive signal) were determined along a line drawn on the confocal images using ImageJ software. Error bars indicate standard deviation (SD) (*n* = 50 images). Scale bars = 10 μm. (**G**) The mutation of serine-67 to glutamic acid reduced the interaction between PRE1 and IBH1 in Arabidopsis. Plants expressing IBH1-Myc alone, the co-expression IBH1-Myc/PRE1-YFP, or co-expression IBH1-Myc/PRE1^S67E^-YFP were used for the coimmunoprecipitation (CoIP) assays using GFP-Trap magnetic beads and were immunoblotted using anti-Myc and anti-YFP antibodies.

**Figure 4 ijms-21-09183-f004:**
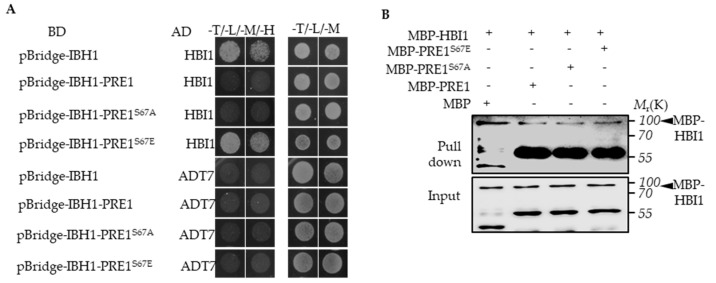
Phospho-mimicking mutation of Ser-67 to Glu-67 resulted in mPRE1 losing its ability to prevent the interaction between IBH1 and HBI1. (**A**) A Yeast Three-Hybrid (Y3H) assay is performed in which PRE1 or mPRE1 and IBH1 are expressed together with a pBridge^TM^ vector equivalent to pGCBDT7 (BD) and HBI1 are connected to an pGCADT7 (AD)vector and co-transformed to yeast-competent cells. The yeast strains were cultured on a –Trp/-Leu/-Met SD medium in order to select the clones that co-express IBH1, PRE1s, and HBI1, then were transferred to –Trp/-Leu/-Met/-His SD medium to test the effects of PRE1s on the interactions between IBH1 and HBI1. (**B**) The in vitro pull-down assays show that the GST-IBH1 binding ability to MBP-HBI1 is inhibited by MBP-PRE1 and MBP-PRE1^S67A^ but not MBP alone or MBP-PRE1^S67E^. The anti-MBP antibody was used to show the protein levels of the Western blot. The arrowheads indicate the MBP-HBI1 proteins.

**Figure 5 ijms-21-09183-f005:**
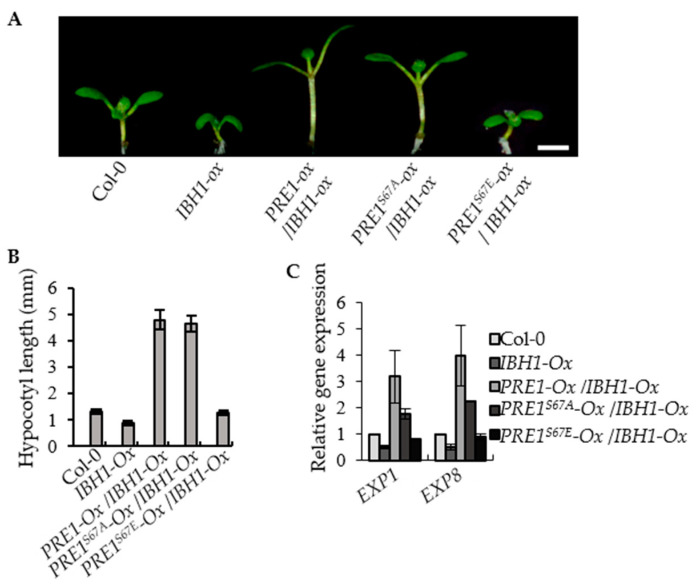
PRE1^S67E^ failed to suppress the dwarf phenotypes of *IBH1-Ox*. (**A**) Overexpression of *PRE1* and *PRE1^S67A^* but not *PRE1^S67E^* suppressed the dwarf phenotypes of *IBH1-Ox*. The *IBH1-Ox* and the F1 plant of *IBH1-Ox* crossing with *PRE1-Ox*, *PRE1^S67A^-Ox,* and *PRE1^S67E^-Ox* grown on a half-strength MS medium for 7 days. Scale bar = 2 mm. (**B**) The average hypocotyl lengths of the Col-0, *IBH1-Ox, IBH1-Ox/PRE1-Ox, IBH1-Ox/PRE1^S67A^-Ox,* and *IBH1-Ox/PRE1^S67E^-Ox* measured from at least 20 plants. Error bars indicate the standard deviation (SD; *n* = 3). (**C**) Quantitative RT-PCR analyses of the gene expression of *EXP1* and *EXP8* in Col-0, *IBH1-Ox, IBH1-Ox/PRE1-Ox*, *IBH1-Ox/PRE1^S67A^-Ox*, and *IBH1-Ox/PRE1^S67E^-Ox* plants. *PP2A* was used as the internal control. Error bars are the SD from three biologic replicates.

**Figure 6 ijms-21-09183-f006:**
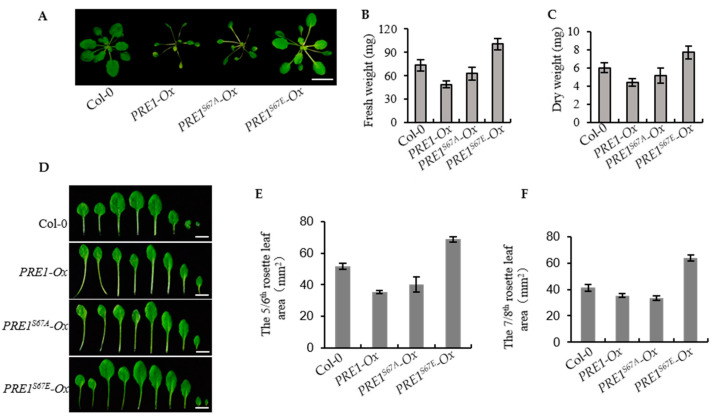
Overexpression of *PRE1^S67E^* increases the biomass of the plant. (**A**) Representative plants of the wild type and the overexpression of the *PRE1, PRE1^S67A^,* and *PRE1^S67E^* transgenic lines grown in soil for 3 weeks. Scale bar = 10 mm. (B,C) The overexpression of *PRE1^S67E^* significantly increased the biomass in Arabidopsis. (**B**) The fresh weight and (**C**) dry weight of the 3-week-old wild type and *PRE1, PRE1^S67A^,* and *PRE1^S67E^* overexpression transgenic lines were measured and analyzed. Error bars indicate the standard deviation (SD; *n* = 3). (**D**) Overexpression of *PRE1* and *PRE1^S67A^* displays elongated petioles, light-green, and narrow leaves, while the overexpression of *PRE1^S67E^* shows the slightly longer petioles, green, and round leaves compared with that of the wild type. Rosette leaves of the wild type and the overexpression of the *PRE1, PRE1^S67A^, and PRE1^S67E^* transgenic lines come from plants grown in soil for 3 weeks. Scale bars = 5 mm. The areas of the (**E**) 5/6th and (**F**) 7/8th rosette leaves of the wild type and the overexpression of the *PRE1, PRE1^S67A^,* and *PRE1^S67E^* transgenic lines measured from at least 15 plants. Error bars indicate standard deviation (SD; *n* = 3).

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
