# Peer review of "Phospho-Mutant Activity Assays Provide Evidence for the Negative Regulation of Transcriptional Regulator PRE1 by Phosphorylation"

_ijms, 2020, doi:10.3390/ijms21239183_

Round 1
Reviewer 1 Report
Manuscript in addition to poor english and style, is full of thought leaps and mistakes. It looks that authors submitted manuscript too early without detailed correction.
Examples:
line 12 proteins that acts
line 23 had no strongly effects
63-PREs also reported
64 ...proteins were further to heterodimerize..
72 reported to regulated
73 are light-induced phosphorylated by PHOTORE GULATORY...
81,82 -what does mean movement...not transport?
98,101,113 and so on.... -to further
107 figure instead of Figure
114 no explanation what exactly means PRE1-Ox and other -ox in entire manuscript!
119 EXP not introduced
Fig. 1 low quality of 1A and 1C-table is not easy to understand what is presented in upper and what in lower part of the table
154 Fig 2 legend and many other Figures is Phospho-mimicing instead of mimicking
163-166 thought leap -from PRE1 suddenly to PRE2 without any explanation
line 176 Figure S4B surely shows interaction of PRE1_S67E with PAR2, so your statement is not true!
Fig. 3A,B lack of explanation what is ADT7 in figure legend
Fig. 3C white backround! no marker with molecular mass
Fig. 3E as control I would expect IBH1-cYFP with nYFP
line 200 to prevent its inhibition of...?
line202 - mutation of PRE1 regulates...I am not sure if mutation as term regulates anything...style...
204 vector consistent..?
Figure4 poor explanation in legend
Figure S2 I do not believe that plot presented on Fig B is consistent with Fig A - IBH1 interaction with PRE2S6E seems to be much stronger in comparison to IBH1 interaction with ADT7, while plot presents near equal results;
Figure S2 C experiment should be performed for PRE2-nYFP with IBH1-cYFP not cCFP like is presented on the figure;
Author Response
Response to Reviewer 1 Comments
Manuscript in addition to poor english and style, is full of thought leaps and mistakes. It looks that authors submitted manuscript too early without detailed correction.
Examples:
line 12 proteins that acts
Response: Thanks for pointing this, we are very sorry for that and we have changed it.
line 23 had no strongly effects
Response: Thanks for pointing this, we have changed it.
63-PREs also reported
Response: Thank you for pointing this, we have changed it.
64 ...proteins were further to heterodimerize..
Response: Thank you for pointing this, we have modified it.
72 reported to regulated
Response: We have changed it.
73 are light-induced phosphorylated by PHOTORE GULATORY...
Response: We have changed it.
81,82 -what does mean movement...not transport?
Response: We have changed ‘movement’ to ‘transport’ according to the reference [49].
98,101,113 and so on.... -to further
Response: We have revised it with respect to the article logic.
107 figure instead of Figure
Response: Thank you for pointing this, we have changed it.
114 no explanation what exactly means PRE1-Ox and other -ox in entire manuscript!
Response: Thank you for pointing this, we have added the explanation of ‘-Ox’ in the firstmention about it in our manuscript.
Overexpression of PRE1S67A (PRE1S67A-Ox) showed the increased cell elongation and light green, narrow leaves, which are similar with that of PRE1 overexpression (PRE1-Ox) transgenic plants, while overexpression of PRE1S67E (PRE1S67E-Ox) displayed the slightly increased cell elongation.
119 EXP not introduced
Response: Thank you for pointing this, we have added the introduction of ‘EXP’ and the related reference [58].
EXPANSIN1 (EXP1) and EXPANSIN8 (EXP8) are two genes encoding the cell wall proteins which loosen the cell wall [58].
Fig. 1 low quality of 1A and 1C-table is not easy to understand what is presented in upper and what in lower part of the table
Response: We have improved the quality of Figure 1 and added more details in the article and figure legent.
We then generated transgenic Arabidopsis plants expressing PRE1, PRE1S46A, PRE1S46E, PRE1S67A, and PRE1S67A driven by the constitutive 35S promoter. Nearly 100 T1 plants for each construct were used to analyze the growth phenotypes. According to the petiole length of the fifth leaf, we classify the transgenic plants into four categories: similar to wild type (no phenotype), weak phenotype, middle phenotype and strong phenotype. As shown in the figure 1B and 1C, mutations Ser-46 to Ala-46 or Glu-46 have no effects on the phenotypes that by overexpression PRE1, indicating that the Ser-46 of PRE1 is not important for the PRE1 function. And the phenotype of mutation Ser-67 to Ala-67 is similar with that of PRE1 overexpression. However, overexpression the PRE1S67E reduced the ratio of plants showing long petiole phenotypes and increased ratio with no phenotype. These results indicated that the phosphorylation at Ser-67 is very important to the function of PRE1
The number (the upper parts of table) and percentage (the bottom parts of table) of each category of phenotypic severity among the T1 transgenic plant populations with overexpression of PRE1 or mutant versions of PRE1.
154 Fig 2 legend and many other Figures is Phospho-mimicing instead of mimicking
Response: We have checked all the descriptions used ‘Phospho-mimicing’.
163-166 thought leap -from PRE1 suddenly to PRE2 without any explanation
Response: Thank you for pointing this, we have added the explanation of PRE2 [34].
PRE2 is a homolog of PRE1 and regulates gibberellin-dependent responses in Arabidopsis thaliana [34]. Protein sequences analysis indicated that the Ser-68 in PRE2 is equivalent to the conserved residue Ser-67 in PRE1 protein (Figure 1A).
line 176 Figure S4B surely shows interaction of PRE1_S67E with PAR2, so your statement is not true!
Response: Thank you for pointing this, we have rewritten the statement.
Fig. 3A,B lack of explanation what is ADT7 in figure legend
Response: Thank you for pointing this, we have added the explanation.
ADT7 means the GAL4AD protein that is used for negative control.
Fig. 3C white backround! no marker with molecular mass
Response: Thank you for pointing this out, we have added the protein markers in these images.
Fig. 3E as control I would expect IBH1-cYFP with nYFP
Response: Sorry for this misleading! In this expeiment, we used the nYFP as control.
line 200 to prevent its inhibition of...?
Response: Thanks for for pointing this out, we have revised the sentence.
Previous studies showed that PRE1 interact with IBH1 to prevent the interaction between IBH1 and HBI1, which directly induces the expression of genes encoding cell wall-loosening enzymes [32].
line202 - mutation of PRE1 regulates...I am not sure if mutation as term regulates anything...style...
Response: Thanks for your suggestion, we have changed it.
204 vector consistent..?
Response: Thank you for pointing this out, we have changed it.
To test whether PRE1 S67E regulates the interaction between IBH1 and HBI1, we performed the Yeast Three-Hybrid (Y3H) assay.
Figure4 poor explanation in legend
Response: As you suggested, we have changed it.
Figure 4. Phospho-mimicing mutation of Ser-67 to Glu-67 resulted in the mPRE1 losing the ability to prevent the interaction between IBH1 and HBI1. (A) Yeast Three-Hybrid (Y3H) assay was performed in which PRE1 or mPRE1 and IBH1 are expressed together by a pbridgeTM vector equivalent to BD and HBI1 are connected AD vector and co-transformed to yeast competent cells. Yeast strains were cultured on –Trp/-Leu/-Met SD medium to select the clones co-express IBH1, PRE1s and HBI1, then transferred to –Trp/-Leu/-Met/-His SD medium to test the effect of PRE1s on the interaction between IBH1 and HBI1.(B) In vitro pull-down assays showed GST-IBH1 binding ability to MBP-HBI1 was inhibited by MBP-PRE1 and MBP-PRE1S67A but not MBP alone or MBP-PRE1S67E. Anti-MBP antibody was used to show the proteins level of western blot. The arrowheads indicate the MBP-HBI1 proteins.
Figure S2 I do not believe that plot presented on Fig B is consistent with Fig A - IBH1 interaction with PRE2S6E seems to be much stronger in comparison to IBH1 interaction with ADT7, while plot presents near equal results;
Response: Thanks for your careful reading of our manuscript. The yeast with indicated vectors were grown on Leu-/Trp-/His- dropout solid medium for 3 days under 30℃, so it looks like that the yeast strains presented on FigureS2A were much stronger due to their overgrowth. However, the histogram of FigureS2B was obtained for an β-gal analysis, which was performed by liquid assay using ONPG. Forβ-gal analysis, we picked 5 individual yeast colonies from each condition that were grown on Leu-/Trp- dropout solid medium. Then, we cultured these individual yeast colonies overnight at 30 °C in 5 ml of indicate selective medium until to OD600=0.5-1.0. Next, we obtained the β-gal activity following the indicated method. Hence, differences between FigureS2A and FigureS2B may due to the two different methods. Even so, I believe that the interaction between PRE2S68E and IBH1 is weaker than the interaction between IBH1 and PRE2.
Figure S2 C experiment should be performed for PRE2-nYFP with IBH1-cYFP not cCFP like is presented on the figure;
Response: Sorry for the misleading, we have corrected it in the new version of our manuscript.

Reviewer 2 Report
The manuscript "Phospho-Mutant Activity Assays Provide Evidence for the Negative Regulation of Transcriptional Regulator PRE1 by Phosphorylation." by Wang et al. analyses the effects of the phosphomimetic mutations of PRE1.
1.) The abstract and parts of the introduction are not easy to understand, because IBH1 and HBI1 are not clearly introduced or explained and therefore the potential effect of PRE1 phosphorylation is not clear.
2.) Please add the molecular weight marker to the Western Blots.
3.) Please include the supplementary data with the required uncropped Western Blots.
4.) Please explain the statistical analysis in more detail in the methods part (e.g. which p value was used to analyze significant differences ?).
5.) Extensive editing of English language and style required.
Author Response
Response to Reviewer 2 Comments
The manuscript "Phospho-Mutant Activity Assays Provide Evidence for the Negative Regulation of Transcriptional Regulator PRE1 by Phosphorylation." by Wang et al. analyses the effects of the phosphomimetic mutations of PRE1.
1.) The abstract and parts of the introduction are not easy to understand, because IBH1 and HBI1 are not clearly introduced or explained and therefore the potential effect of PRE1 phosphorylation is not clear.
Response 1: Thanks for your thoughtful comments. As you suggested, we have added the introduction of IBH1 and HBI1 in the new version of manuscript.
Abstract: PRE1 as a positive regulator of cell elongation activates the HBI1’s DNA binding by sequestering its inhibitor IBH1.
Introduction:HBI1, as an bHLH protein, is a positive regulator in regulating cell elongation, while IBH1 is a negative regulator of cell elongation. HBI1 directly binds to the promoters of downstream genes to regulate their expression, and IBH1 interacts with HBI1 to inhibit the DNA binding ability of HBI1, whereas PRE1 interacts with IBH1 to prevent its inhibition of HBI1 [32].
2.) Please add the molecular weight marker to the Western Blots.
Response 2: Thank you for your valuable advice. We have added the molecular weight marker to all the western blots.
3.) Please include the supplementary data with the required uncropped Western Blots.
Response 3: Thank you for pointing this out, we have provided the uncropped western blot images in Figure S6.
4.) Please explain the statistical analysis in more detail in the methods part (e.g. which p value was used to analyze significant differences ?).
Response 4: Thanks for your suggestion, we have rewritten the statistical analysis in the methods part in the new version of our manuscript.
5.) Extensive editing of English language and style required.
Response 5: Thank you for pointing this out. We have extensively edited our manuscript.

Round 2
Reviewer 1 Report
The manuscript is better, however still not all comments were concerned fully. Aditionally I was giving just examples of errors, so authors should perform additional check-up of text.
From major points:
Authors gave no responce to my comments concerning too white background of western blots images. In addition, new included Figure S6 also contains images with totally white background, which is totally not proper way of western blot presentation while it can influence the strenght and quantity of protein bands!
Fig.1 A and C quality is still poor
Minor points:
-In Figures S1 and S4 legends still mimicing exists!
-line 40 still no space before [17-19]
-line 275 On the contrary?
-line 278 what was interesting was?
-line 383 beta-. gal?
Figures S1 and S4 still mimicing in legends exists !!!
Explanations concerning S2 Figure should be also at least partially included in manuscript.
Author Response
Comments and Suggestions for Authors
The manuscript is better, however still not all comments were concerned fully. Aditionally I was giving just examples of errors, so authors should perform additional check-up of text.
From major points:
Authors gave no responce to my comments concerning too white background of western blots images. In addition, new included Figure S6 also contains images with totally white background, which is totally not proper way of western blot presentation while it can influence the strenght and quantity of protein bands!
Response: Thanks for pointing this. We have adjusted the contrast of the images of western blot to show the gray background.
Fig.1 A and C quality is still poor
Response: Thanks for pointing this, we have changed them.
Minor points:
-In Figures S1 and S4 legends still mimicing exists!
Response: Thanks for pointing this, we have changed it.
-line 40 still no space before [17-19]
Response: Thanks for pointing this, we have changed it.
-line 275 On the contrary?
Response: Thanks for pointing this, we have changed it.
-line 278 what was interesting was?
Response: Thanks for pointing this, we have changed it.
-line 383 beta-. gal?
Response: Thanks for pointing this, we have changed it.
Figures S1 and S4 still mimicing in legends exists !!!
Response: Sorry for this, we have changed them.
Explanations concerning S2 Figure should be also at least partially included in manuscript.
Response: Thank you for pointing this out, we have introduced the Figure S2 in the new version of manuscript.
Reviewer 2 Report
The authors have addressed all of my points.
Author Response
Comments and Suggestions for Authors
The authors have addressed all of my points.
Response: Thank you very much for your positive comment.